# Socio-Economic and Demographic Factors Associated with COVID-19 Mortality in European Regions: Spatial Econometric Analysis

Mateusz Szysz [1,*] and Andrzej Torój [2]

1 Department of Mathematical Economics, SGH Warsaw School of Economics, Niepodległości 162, 02-554 Warsaw, Poland
2 Institute of Econometrics, SGH Warsaw School of Economics, Niepodległości 162, 02-554 Warsaw, Poland
* Correspondence: mateuszsysz98@gmail.com

**Abstract:** In some NUTS 2 (Nomenclature of Territorial Units for Statistics) regions of Europe, the COVID-19 pandemic has triggered an increase in mortality by several dozen percent and only a few percent in others. Based on the data on 189 regions from 19 European countries, we identified factors responsible for these differences, both intra- and internationally. Due to the spatial nature of the virus diffusion and to account for unobservable country-level and sub-national characteristics, we used spatial econometric tools to estimate two types of models, explaining (i) the number of cases per 10,000 inhabitants and (ii) the percentage increase in the number of deaths compared to the 2016–2019 average in individual regions (mostly NUTS 2) in 2020. We used two weight matrices simultaneously, accounting for both types of spatial autocorrelation: linked to geographical proximity and adherence to the same country. For the feature selection, we used Bayesian Model Averaging. The number of reported cases is negatively correlated with the share of risk groups in the population (60+ years old, older people reporting chronic lower respiratory disease, and high blood pressure) and the level of society's belief that the positive health effects of restrictions outweighed the economic losses. Furthermore, it positively correlated with GDP per capita (PPS) and the percentage of people employed in the industry. On the contrary, the mortality (per number of infections) has been limited through high-quality healthcare. Additionally, we noticed that the later the pandemic first hit a region, the lower the death toll there was, even controlling for the number of infections.

**Keywords:** COVID-19; spatial analysis; mixed-W model; mortality; morbidity

**JEL Classification:** C29; I10



## 1. Introduction

Severe acute respiratory syndrome coronavirus 2 (SARS-CoV-2), which causes a cluster of respiratory tract infections called coronavirus disease 2019 (COVID-19), was first identified and characterized at the end of 2019 in China (Zhou et al. 2020). In the early months of 2020, the virus spread worldwide; thus, on March 11, the WHO assessed that COVID-19 could be characterized as a pandemic (World Health Organization 2020a). In Europe, the first cases were reported in Bordeaux, France, on 24 January 2020, (Stoecklin et al. 2020). Several weeks later, on March 13, WHO remarked that Europe had become an epicenter of the pandemic with more reported cases and deaths than the rest of the world combined, apart from China (World Health Organization 2020b). In total, between March and December 2020, the number of excess deaths in the EU compared with the 2016–2019 average reached more than half a million (Eurostat 2020j).

However, the pandemic did not hit individual European regions equally. There were considerable differences in morbidity and mortality both between countries (Bartscher et al. 2021; Coccia 2021; Fu et al. 2021) and regions in the same country as in Italy (Bourdin et al.

2021), in Spain (Cacho et al. 2020), or in the US (Mollalo et al. 2020). The variation in the outcomes of the pandemic might be caused by multiple factors related to the economy (Goutte et al. 2020; Kong et al. 2021), demography (Hradsky and Komarek 2021; Sorci et al. 2020), genetic predispositions (Schillaci 2020; Yamamoto et al. 2021), living environment (Coccia 2021; Nazir et al. 2021; Yao et al. 2020), etc. Moreover, the adopted state's policy concerning short-term restrictions and people's attitudes towards them (Amdaoud et al. 2021; Xie et al. 2021) or long-term activity aimed at improving the quality of the health care system (Yalaman et al. 2021) were other potential factors.

Multiple studies highlighted the determinant effect of age and the presence of co-morbidities on mortality rate (Elezkurtaj et al. 2021; Imam et al. 2020; Mueller et al. 2020; O'Driscoll et al. 2021; Poblador-Plou et al. 2020). According to Bonanad et al. (2020), among patients aged less than 50 mortality rate was lower than 1.1% and increased exponentially after this age. The most significant increase in mortality risk compared with the immediately younger age group was observed in patients aged 60–69 (compared to 50–59). Among diseases associated with a higher risk of infection and death were respiratory diseases, diabetes, and hypertension (Biswas et al. 2020; Fang et al. 2020). Moreover, some studies suggested a correlation between the growth rate of SARS-CoV-2 infection and a genetic profile, particularly the distribution of the Y-chromosome haplogroup R1b (Schillaci 2020).

Large crowds of people may facilitate the transmission of the virus. Brown et al. (2021) showed that shared bedrooms and bathrooms in nursing homes were associated with more extensive and deadlier COVID-19 outbreaks. Similar effects might be found in indoor settings such as large and crowded work establishments. Furthermore, people living in densely populated agglomerations with many public places might be more vulnerable to infection. Indeed, some studies indicated that population density was correlated with the basic reproductive number ($R_0$) of SARS-CoV-2 (Sy et al. 2021). On the contrary, Carozzi et al. (2022) showed that the density did not directly affect the number of cases or fatalities but the timing of the outbreaks.

Another factor that might explain the differences in the number of infections and fatalities between regions was the state's policy concerning diagnostics, testing, and treatments. Broad access to resources such as hospitals, medical staff, and high healthcare expenditures might reduce fatality rates (Amdaoud et al. 2021; Coccia 2021; Oshinubi et al. 2021). However, in the short run, it was not possible to sufficiently increase these resources. Therefore, governments imposed restrictions aimed at decreasing the number of infections. According to Oh et al. (2021), reductions of up to 40% in commuting mobility was associated with a reduced number of cases, especially early in the pandemic, in the majority of examined 36 (mainly OECD) countries. In the US, closing schools and colleges between March and May 2020 had a similar negative effect on COVID-19 incidence and fatality rates (Auger et al. 2020).

Some studies indicated the importance of environmental factors on COVID-19 morbidity and mortality. For instance, Bourdrel et al. (2021) suggested that air pollution might be linked to an increase in COVID-19 severity through its impact on chronic disease arising from decreased immune response and thus facilitating replication of the virus and penetration of host cells. On the contrary, Xu et al. (2021) showed that changes in air pollution alone were insufficient to contain the spread of SARS-CoV-2, with other factors such as warmer temperatures and moderate outdoor ultraviolet exposure having greater effects.

Enforcement of the restrictions is as important as their initial imposition. Thereupon societies characterized by the high social trust may have a larger propensity to fulfill obligations and listen to medical experts' advice. Indeed, as shown by Amdaoud et al. (2021), social trust was one of the key factors in reducing the fatality rate during the coronavirus pandemic. Presumably, trust in authorities may be insufficient when a large part of the society does not share the same values as policymakers. It may happen when most people are more concerned about economic damage and a breach of personal freedom than the health consequences of infection (Szysz 2022).

The number of cases and deaths in neighboring regions were critical factors in predicting the outcome of the pandemic, and the role of this factor is growing with smaller territorial units becoming observable to the econometrician. Multiple studies conducted on different levels of regional aggregation considered such a form of possible dependence by employing spatial econometric models. In the US, the analyzes were performed, for instance, at the level of counties (Kandula and Shaman 2021; Sun et al. 2020), whereas in Europe at the level of states (Sannigrahi et al. 2020), NUTS (Nomenclature of Territorial Units for Statistics) regions (Amdaoud et al. 2021), or administrative units within a specific country (Ehlert 2021). Amdaoud et al. (2021) used the SAR model to analyze socioeconomic factors that impacted the levels of the COVID-19 death rate in different EU regions. On the other hand, Ehlert (2021) studied the association between different socioeconomic variables and COVID-19-related cases and deaths in German counties with SAR, SEM, and SAC models. We decided to explore these approaches by employing the SARAR model, which is a generalization of the SAR and SEM models, to analyze the numbers of COVID-19 cases and deaths in European regions.

Analyzing smaller regions may be preferred due to more insightful knowledge about differences within larger units. Moreover, a higher level of granulation allows for the production of a larger number of observations and, thus, more robust statistical analysis results. On the other hand, many variables are not available at a sufficiently small level of aggregation. Using variables from different aggregation levels simultaneously can be proposed as a compromise. In the analysis of NUTS regions, such an approach would consist of using country averages for variables unavailable at the lower level of aggregation. However, this mismatch may lead to spurious spatial autocorrelation of errors in the model.

One of the possible solutions to the problem is the so-called multiparameter, or mixed-W approach, based on using multiple spatial adjacency matrices (Hays et al. 2009). We propose the following specification:

$$
\begin{aligned}
\boldsymbol{y} &= \boldsymbol{X}\boldsymbol{\beta} + \rho \boldsymbol{W_1}\boldsymbol{y} + \boldsymbol{\epsilon} \\
\boldsymbol{\epsilon} &= \lambda_1 \boldsymbol{W_1}\boldsymbol{\epsilon} + \lambda_2 \boldsymbol{W_2}\boldsymbol{\epsilon} + \boldsymbol{u}
\end{aligned}
\tag{1}
$$

In this setup, $\boldsymbol{W_1}$ is a typical neighboring matrix, for example, based on proximity, and accounts for the spatial spread of COVID-19. $\boldsymbol{W_2}$, in turn, is a matrix indicating adherence to the same country and represents the possibly omitted country-specific factors that, in a single-country study such as Ehlert (2021), could have been ignored. It also controls for artificial spatial autocorrelation resulting from applying country-level regressors to sub-national regions. The mixed-W approach was, for instance, used by Lasoń and Torój (2019) in the 2015 Polish parliamentary election study where $\boldsymbol{W_2}$ indicated adherence of two poviats to the same constituency, which accounts for the unobservable personal qualities of the candidates that the voters in that particular constituency are facing.

In our study, we use a similar idea (details of the model specification were described in the next section). The data covered 189 (mainly NUTS 2) regions from 19 EU countries. Some explanatory variables were aggregated at the regional level and others at the country level (see Table 1 for details). Then, we use a mixed-W model to accommodate the spatial correlation effects stemming from the use of country-level characteristics. One can think of this empirical strategy as an alternative to a spatial multi-level model.

All in all, the study aimed to identify socio-economic and demographic factors responsible for acute differences in the increased number of deaths between the European regions, in a multi-country setup, with spatial regression models that take two types of spatial effects into account: (i) naturally arising from spatial interactions and (ii) technically arising due to inclusion of country-level characteristics that substantially extend the set of region-level characteristics. We estimated two types of spatial regression models explaining both the number of reported COVID-19 cases per 10,000 inhabitants and the percentage increase in deaths compared to the 2016–2019 average. As the study was based on cross-sectional

data, we focused only on 2020 to eliminate the effect of different dynamics of vaccination between countries.

This paper is organized as follows: firstly, we discussed the dataset with particular emphasis on the level of aggregation of individual explanatory variables. Secondly, we described methods used for feature selection and estimating models. Finally, we presented and discussed the obtained results.

**Table 1.** List of the explanatory variables included in the analysis with data source and the level of aggregation.

| Variable | Definition | Level of Aggregation | Year | Source |
|---|---|---|---|---|
| Population aged 60 | percentage of population aged 60 or over | NUTS 2 (except NUTS 1 in DE) | 2020 * | Eurostat (2020g) |
| Health expenditure | total health care expenditure per inhabitant in PPS [$] | country | 2019 | Eurostat (2020c) |
| Doctors | number of medical doctors per hundred thousand inhabitants | NUTS 2 (except NUTS 1 in DE) | 2014–2019 | Eurostat (2020d) |
| GDP | GDP per inhabitant in PPS as a percentage of the EU27 average | NUTS 2 (except NUTS 1 in DE) | 2019 | Eurostat (2020h) |
| Density | annual average number of persons per square kilometer | NUTS 2 (except NUTS 1 in DE) | 2019 | Eurostat (2020f) |
| Day | number of days between the beginning of 2020 and the day when number of COVID-19 cases per 10,000 inhabitants exceeded 1 | NUTS 2 (except NUTS 1 in DE) | 2020 | COVID-19 European regional tracker (Naqvi 2021) |
| Pollution | average exposure to air pollution by fine particulate matter ($PM_{2.5}$) [µg/m$^3$ per year] | NUTS 2 (except NUTS 1 in DE) | 2017 | Eurostat (2020b) |
| NACE B-E | percentage of people employed in NACE B, C, D, E sectors | NUTS 2 (except NUTS 1 in DE) | 2020 * | Eurostat (2020a) |
| Mean stringency | average value of the COVID-19 stringency index between 28 days before the day when the number of cases per 10,000 exceeded 1 and the end of the year | country | 2020 | Our World in Data (2020) |
| R1b | percentage of population with R1b haplogroup | country | 2017 | Eupedia (2020) |
| Diabetes | percentage of people aged 65 or over reporting diabetes | country | 2014 | Eurostat (2020e) |
| Chronic lower respiratory disease | percentage of people aged 65 or over reporting chronic lower respiratory disease | country | 2014 | Eurostat (2020e) |
| High blood pressure | percentage of people aged 65 or over reporting high blood pressure | country | 2014 | Eurostat (2020e) |
| Economic damage | percentage of population strongly claiming that regarding the consequences of the restriction measures the health benefits are greater than the economic damage | country | 2020 | Kantar for the European Parliament (2020) |
| Trust in health authorities | percentage of population trusting most health authorities to inform them about the coronavirus pandemic | country | 2020 | Kantar for the European Parliament (2020) |
| Trust in government | percentage of population trusting most government to inform them about the coronavirus pandemic | country | 2020 | Kantar for the European Parliament (2020) |

\* The data is on 1 January 2020.

## 2. Materials and Methods

### 2.1. Dataset

The analysis covers 16 NUTS 1 German regions and 173 NUTS 2 regions from 18 different EU countries (see color-coded countries in Figures 1 and 2). We could not use lower-level units for Germany because NUTS 2 data concerning the number of deaths was unavailable. Moreover, we excluded the following regions due to missing data for some essential explanatory variables: the Canary Islands (ES70), Melilla (ES64), the Azores (PT20), Madeira (PT30), and Aland (FI20). We estimated two types of models explaining the relative number of cases and the percentage of deaths increase. We emphasize that the latter may include both deaths from COVID-19 and deaths related to the pandemic, which resulted, for instance, from limited access to health care. Using a broader category eliminated the impact of differences in the methodology of classifying deaths between countries.

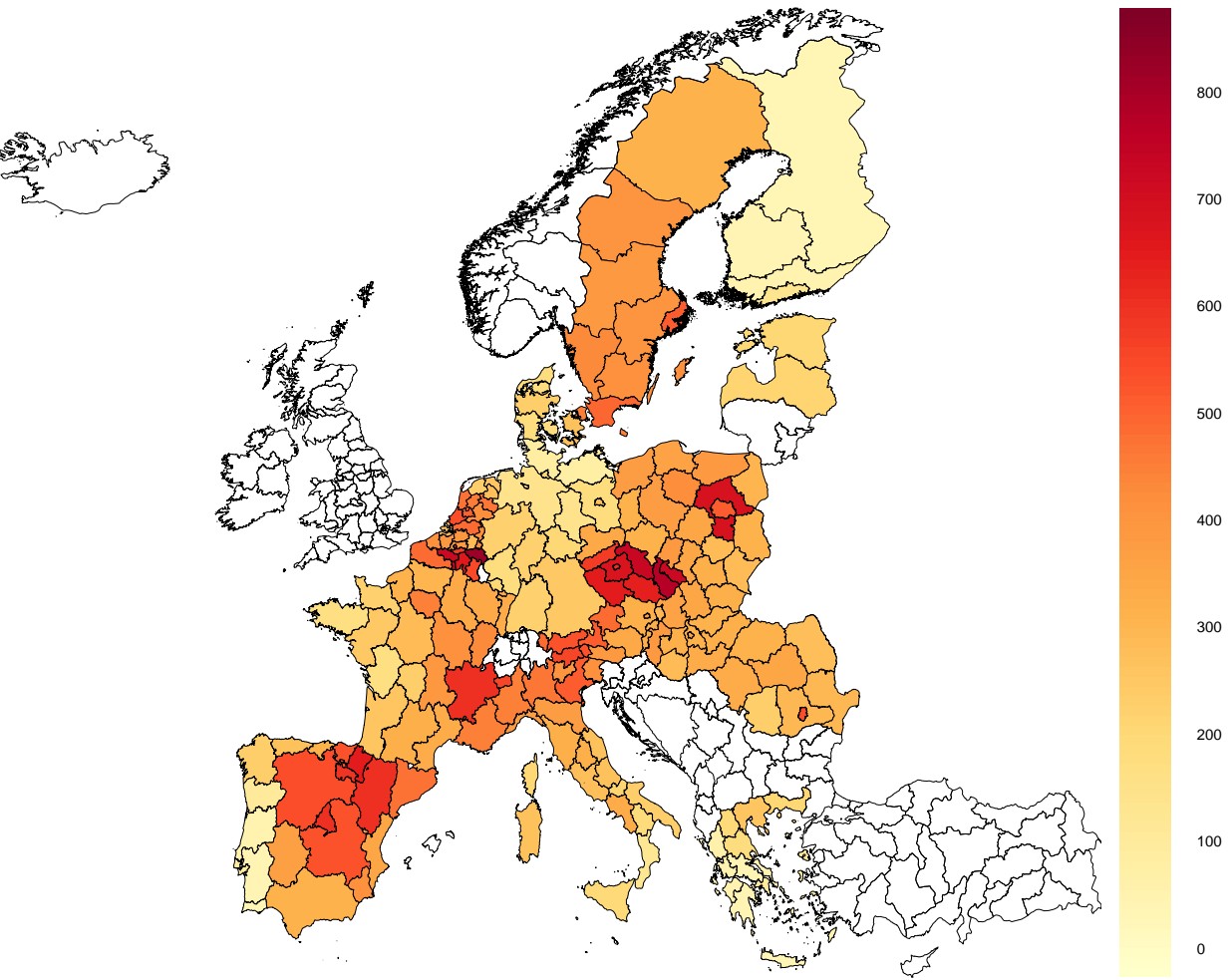

**Figure 1.** Number of reported COVID-19 cases per 10,000 inhabitants in 2020 in selected EU regions.

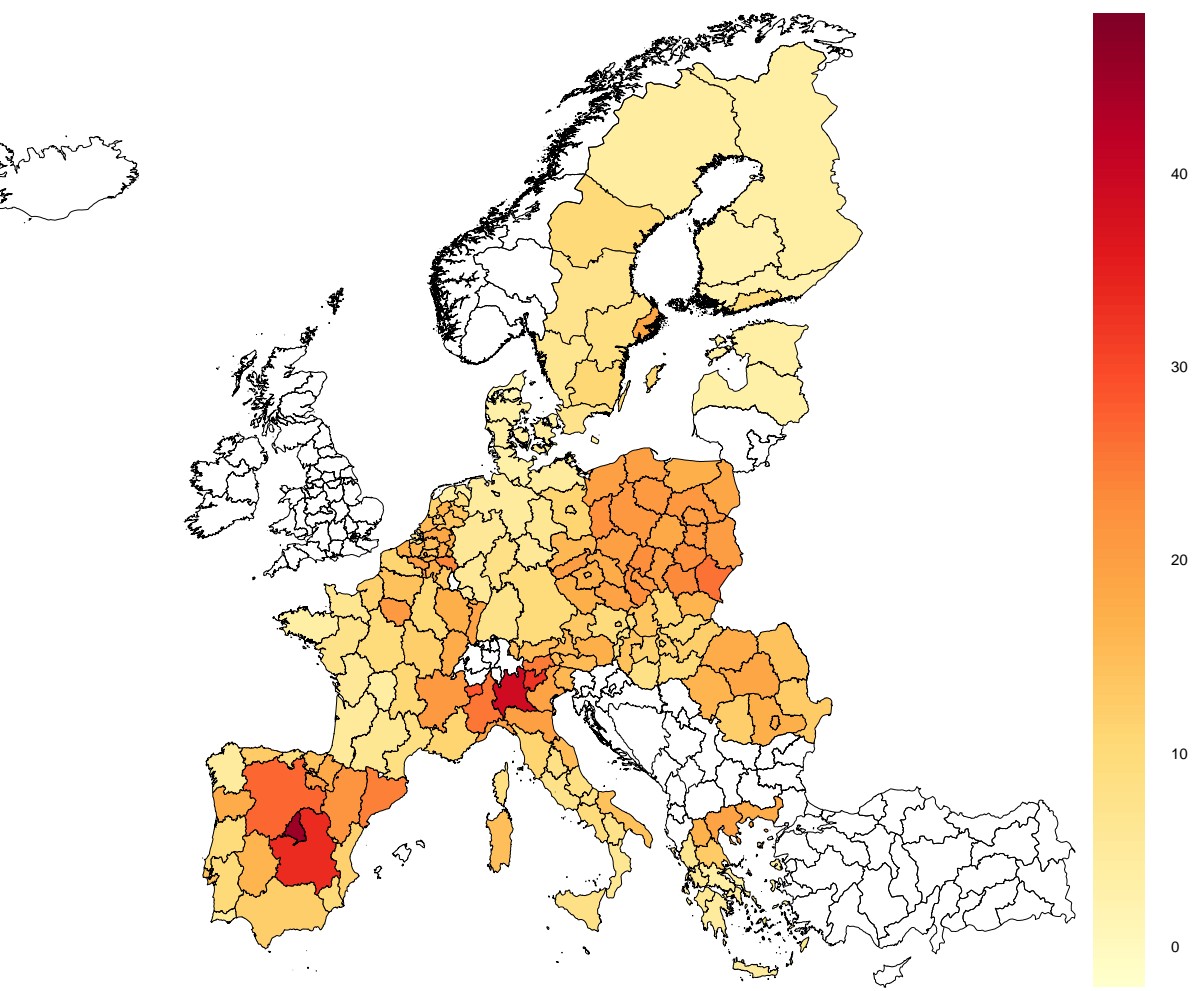

**Figure 2.** Percentage increase in the number of deaths in selected EU regions in 2020 compared with the average number of deaths between 2016 and 2019.

The endogenous variables were defined as follows:

- Relative number of cases—the cumulative number of infections per 10,000 people as of 31 December 2020 (except for Portuguese regions for which the measurements were cumulated until December 28 due to missing data);
- Deaths increase—percentage increase in the number of deaths in 2020 compared to the 2016–2019 average. To make sure that the results are robust with respect to the adopted working definition of excess mortality (see Kepp et al. (2022) for a related discussion), we also look at a deviation of the number of deaths from the 2016–2019, region-specific trend. This has virtually no impact on the results (details available upon request).

The former was also one of the explanatory variables for the latter. It was based on the data from the COVID-19 European regional tracker, a repository containing a daily number of cases and deaths related to COVID-19 in different European regions (Naqvi 2021). Depending on the country, the data was initially provided at NUTS 3 and NUTS 2 levels. However, we aggregated them to the NUTS 2 level (and NUTS 1 for Germany) using population-weighted averages. The result of the procedure with measurements for individual regions was presented in Figure 1. All the studied regions are marked with colors depending on the relative number of reported coronavirus cases in 2020. The other endogenous variable was based on the weekly death statistics provided by Eurostat (Eurostat 2020i). Figure 2 shows its distribution across regions.

All the explanatory variables were aggregated at one of two levels: NUTS 2 (NUTS 1 in the case of Germany) or a country level. Their summary is shown in Table 1. Each predictor was based on 2020 or the most recent available year before 2020.

Diabetes, Chronic lower respiratory disease, and High blood pressure indicate the share of the population reported selected diseases. It is worth noting that each variable is based on the survey data. On the one hand, we can expect these variables to correlate with the actual share of people with chronic diseases that increase the probability of severe coronavirus infection and death. On the other hand, people from these groups could fear most of COVID-19's consequences and be more prone to keep social distance (Heid et al. 2021), which could result in fewer infections. Similar arguments apply to the Population aged 60 variable, which shows the percentage of people aged 60 or over.

Health expenditure and Doctors show the possibility of successful treatment of COVID patients. Thereupon, both of them should be negatively correlated with the deaths increase. However, the larger number of medics and greater expenditure on health care could also lead to better diagnostics and an effective reduction in cases.

The Doctors variable was the only one created based on different years and the only one that required missing data imputation. The procedure details were included in Appendix A.1.

Density and NACE B-E (the definition is provided in Table 1) demonstrate the probability of social contact in the region. A higher density is typical for urban areas with many public places that increase the likelihood of accidental encounters with infected people. On the other hand, professions classified into B, C, D, and E sectors that concern industry without construction are usually characterized by large and frequently crowded establishments. Therefore, we could expect both variables to impact the number of infections positively.

Economic damage, Trust in health authorities, and Trust in government can be recognized as behavioral variables. Societies with more people claiming that health benefits from restrictions outweighed the economic damage might be more prone to impose and comply with stricter restrictions. Similarly, people who trust medical experts may be more inclined to follow their advice and obey the pandemic restrictions. Therefore, we expected these variables to correlate negatively, at least with the number of reported cases. Details on the construction of these variables were included in Appendix A.1.

Day shows how late the pandemic hit the region. A shorter time from the first cases should implicate fewer infections at the end of the year. However, on the other hand, governments of countries hit by the pandemic later had more time to stock on all the necessary resources used in treatment, more effectively healing the infected and reducing the fatality rate.

Moreover, we included R1b, which is a share of the population with the r1b haplogroup, mean stringency based on the average value of the stringency index, and the average exposure to air pollution.

For instance, a larger share of the older population or a larger number of doctors per inhabitant is usually observed in more prosperous regions. Therefore, we included GDP per person as a control variable. Fortunately, as shown in Appendix A.3, it did not cause the multicollinearity problem.

The relations between all the explanatory variables and the dependent ones are shown on scatterplots in Appendix A.2.

### 2.2. Feature Selection

Some variables directly impacted the number of deaths related to COVID-19, and some through a larger number of infections. We used Bayesian Model Averaging implemented in R's BMS package (Zeugner and Feldkircher 2015) to select subsets of variables that might significantly explain each endogenous variable. This method is based on calculating the parameters of multiple candidate models and combining their estimates according to the posterior inclusion probability, which shows how relevant each variable is. The procedure results were shown in Tables 2 and 3.

**Table 2.** Bayesian Model Averaging for the number of infections per 10,000 people.

|  | PIP | Post Mean | Post SD | Cond.Pos.Sign |
|---|---|---|---|---|
| Economic damage | 1.000 | −9.591 | 1.338 | 0.000 |
| NACE B-E | 1.000 | 7.020 | 1.398 | 1.000 |
| High blood pressure | 1.000 | −8.884 | 2.225 | 0.000 |
| Chronic lower respiratory | 1.000 | −22.409 | 4.811 | 0.000 |
| Population aged 60 | 0.997 | −11.782 | 2.965 | 0.000 |
| Day | 0.995 | −1.314 | 0.362 | 0.000 |
| GDP | 0.987 | 1.211 | 0.361 | 1.000 |
| R1b | 0.819 | 1.555 | 0.975 | 1.000 |
| Trust in health authorities | 0.777 | −2.265 | 1.525 | 0.000 |
| Diabetes | 0.475 | 3.856 | 4.966 | 1.000 |
| Mean stringency | 0.391 | 1.304 | 2.049 | 1.000 |
| Trust in government | 0.364 | −1.213 | 2.098 | 0.018 |
| Density | 0.273 | −0.005 | 0.010 | 0.002 |
| Doctors | 0.169 | 0.012 | 0.046 | 0.987 |
| Health expenditure | 0.159 | −0.001 | 0.007 | 0.327 |
| Pollution | 0.153 | −0.098 | 1.002 | 0.432 |

**Table 3.** Bayesian Model Averaging for the death toll increase.

|  | PIP | Post Mean | Post SD | Cond.Pos.Sign |
|---|---|---|---|---|
| Cases | 1.000 | 0.027 | 0.003 | 1.000 |
| Pollution | 0.844 | 0.253 | 0.145 | 1.000 |
| Diabetes | 0.774 | 0.317 | 0.209 | 1.000 |
| Mean stringency | 0.769 | 0.209 | 0.137 | 0.999 |
| Day | 0.587 | −0.020 | 0.020 | 0.000 |
| Doctors | 0.579 | −0.005 | 0.005 | 0.000 |
| Health expenditure | 0.495 | −0.001 | 0.001 | 0.000 |
| Trust in government | 0.383 | −0.094 | 0.145 | 0.000 |
| Trust in health authorities | 0.295 | 0.039 | 0.073 | 0.968 |
| Chronic lower respiratory | 0.261 | 0.118 | 0.252 | 0.909 |
| Population aged 60 | 0.191 | −0.041 | 0.103 | 0.002 |
| Economic damage | 0.121 | 0.010 | 0.034 | 0.994 |
| R1b | 0.119 | 0.004 | 0.017 | 0.861 |
| Density | 0.112 | 0.0001 | 0.0003 | 0.968 |
| GDP | 0.101 | 0.002 | 0.007 | 0.909 |
| NACE B-E | 0.067 | 0.0001 | 0.017 | 0.475 |

In both cases, we decided to estimate models on variables with a posterior inclusion probability greater than 0.4. However, for comparison in the results section, we also presented models estimated on the whole set of predictors.

It is worth noticing that the probability of a positive sign for all the selected variables was greater than 0.99 or smaller than 0.01. It suggests that their signs should remain unaffected by changing the specifications of the models.

### 2.3. Spatial Autocorrelation and Spatial Regression Model

The independence of observations, one of the critical assumptions in linear regression, may not hold in the analysis of infections and deaths related to COVID-19 (Kandula and Shaman 2021). Therefore, we had to test for spatial autocorrelation and regard it when estimating models. As possible patterns for such autocorrelation, we tried three matrices: $W_{borders}$, $W_{distance}$, and $W_{country}$. $W_{borders}$ is a row-standardized matrix based on a queen congruity (two regions are neighbors if and only if their borders share at least one point). $W_{distance}$ is based on the inverse of distances between centers of regions and is also row-standardized. Finally, $W_{country}$ shows the adherence of two regions to the same country. Its $(i, j)$ element is defined as follows:

$$W_{i,j} = \begin{cases} \frac{1}{k(i)-1} & \text{if } i \text{ and } j \text{ adhere to the same country and } i \neq j \\ 0 & \text{otherwise} \end{cases}, \tag{2}$$

where $k(i)$ is the number of regions that belong to a given country. As can be seen, $W_{country}$ is also row-standardized by definition. As indicated in the introductory section, using this matrix has two goals. Firstly, it allows for accounting for unobservable effects at the country level. Secondly, it controls for the artificial autocorrelation resulting from imputing country-level averages instead of region-specific values.

The choice of $W_{borders}$ and $W_{distance}$ as possible patterns for spatial autocorrelation reflects our assumption that the interactions between regions decrease with increasing distance. It is in line with the results of Torój (2022), who showed that most of the traffic in Poland takes place within NUTS 3 regions, not in between them.

To account for the spatial autocorrelation in the model, we used an idea indicated in the introduction. For each type of model, we considered the following specification:

$$\begin{aligned} y &= X\beta + \rho_{borders}W_{borders}y + \epsilon \\ \epsilon &= \lambda_{borders}W_{borders}\epsilon + \lambda_{country}W_{country}\epsilon + u, \end{aligned} \tag{3}$$

where $y$ is the endogenous variable (relative number of cases or deaths increase), $X$ is a set of predictors, and $\epsilon$ is an error term that consists of two spatially autocorrelated parts and an independent part $u$. We referred to this full model as mixed-W SARAR.

We can test several restrictions on this model. For example, if $\lambda_{country} = 0$, we obtain a typical SARAR specification. If $\rho_{borders} = 0$, we obtain a mixed-W SEM model. If both $\rho_{borders} = 0$ and $\lambda_{country} = 0$, we obtain a classic SEM model.

Instead of using $W_{borders}$, we also tried $W_{distance}$, but this choice has led to spatial autocorrelation remaining in the model residuals (see Appendix A.5). Moreover, we investigated another approach based on introducing spatial lags of independent variables to the model (SLX specification). Still, it fit the data much worse than the SARAR model and also did not allow us to account for spatial autocorrelation. It is described in Appendix A.7.

In the study, we used a general-to-specific approach with respect to spatial components, i.e., we started with the full specification shown above and removed spatial components until all of the remaining ones were significant. It should be noted that the set of predictors selected with BMA did not change at any further step. Additionally, we tested for the autocorrelation of models' residuals with Moran's I test (Moran 1950) implemented in R's spdep package (Bivand et al. 2021; Bivand and Wong 2018) with respect to the three matrices mentioned so far.

The parameters of the models were estimated with the maximum likelihood estimator with an assumption of independent and Gaussian errors $u$ and a constant variance. In particular, we transformed Equation (3) to the following form:

$$M \cdot (L \cdot y - X \cdot \beta) = u, \tag{4}$$

where

$$M = I_N - \lambda_{borders} \cdot W_{borders} - \lambda_{country} \cdot W_{country},$$

$$L = I_N - \rho_{borders} \cdot W_{borders},$$

$N$ is the total number of regions, and $I_N$ is a diagonal matrix of size $N$. Finally, we derived the following log-likelihood function formula:

$$\ell\big(\rho_{borders}, \lambda_{borders}, \lambda_{country}, \beta, \sigma\big) = \log(\det(M)) + \log(\det(L)) - \frac{N}{2}\log\big(2\pi\sigma^2\big) - \frac{u^T u}{2\sigma^2}, \tag{5}$$

where

$$u = M \cdot (L \cdot y - X \cdot \beta).$$

The function was maximized with the BFGS algorithm. We used the asymptotic normality property of the maximum likelihood estimator with variance based on Fisher information to obtain the distribution of parameter estimates. Based on that, we tested the significance of each predictor with the Z-test (see the R scripts for details).

## 3. Results

### 3.1. Relative Number of Cases

In 2020, the number of reported COVID-19 cases in the analyzed regions varied from 28.5 cases per 10,000 inhabitants in Algarve, Portugal (PT15), to 823.1 in Liège, Belgium (BE33). Figure 1 shows that the infections were not distributed randomly but rather cluster-wise. Moreover, the differences occurred at the level of countries (e.g., compare Czechia with Finland) and regions in the same country (e.g., compare Mazovia Province with the rest of Poland or central and northern Spain with the rest of the country).

Table 4 summarizes the obtained estimates of coefficients and results of Moran's I test. In column 1, we presented OLS estimates (non-spatial). In columns 2 and 3, the final results of the general-to-specific approach that include the component related to adherence to the same country. Finally, in columns 4 and 5, for comparison, we presented the same models as the previous two but without a component related to adherence to the same country.

In the non-spatial model, Moran's I values are highly significant for each of the neighboring matrices ($p < 10^{-10}$ in all cases), which indicates that accounting for spatial autocorrelation was necessary. The final models are mixed-W SARAR and mixed-W SEM. Differentiating between them depends on the chosen significance level. In the mixed-W SARAR, both $\lambda_{borders}$ and $\lambda_{country}$ are significant at the level of 1% ($p$ equal 0.00229 and 0.00952, respectively), while $\rho_{borders}$ is significant at the level of 10% ($p = 0.0594$). In case of the mixed-W SEM, $\lambda_{borders}$ is significant at the level of 1% ($p < 0.00001$) and $\lambda_{country}$ at the level of 5% ($p = 0.0103$).

Moran's I values for SARAR and SEM models with one spatial adjacency matrix indicate that they do not capture the autocorrelation stemming from the adherence to the same country ($p < 0.001$ in both cases). Therefore, it clearly shows that the mixed-W approach is more appropriate for analyzing the number of cases. It is also worth noting that even though the results of Moran's I test for the non-spatial model indicated a spatial pattern of residuals autocorrelation represented by the matrix based on distance, including the matrix based on shared borders was sufficient to account for this pattern. However, it did not work the other way around, which suggests that the actual unobserved spatial pattern is closer to the one represented by $W_{borders}$. Nevertheless, as indicated previously, in Appendix A.7, we showed the models with $W_{distance}$.

The share of the population aged 60 or over and the share of the elderly suffering from high blood pressure and chronic lower respiratory disease indicate how large part of the population was most at risk of severe COVID-19. Each of these variables was negatively correlated with the number of cases that might be caused by multiple factors. Firstly, older people are usually less mobile than the young, so they are less prone to be infected. Secondly, older people, especially those with comorbidities, could be more afraid of the outcomes of infection, and therefore, they could try to keep a social distance. Finally, some restrictions imposed by governments aimed at protecting the most prone to the severe course of the disease might be at least partially effective. Interestingly, the percentage of diabetics was not significant in any of the models with spatial components. These results align with the explanatory data analysis shown in Appendix A.2.

From behavioral variables, only Economic Damage is significant and, as anticipated, negatively correlated with the number of cases in each specification. It was created based on the questionnaire, and different ways of aggregating the responses were possible (see Appendix A.1). However, the significance of this variable was insensitive to the use of other aggregation methods (see Appendix A.4).

Finally, Table 5 presents standardized regression coefficients, computed as $\frac{SD(x_k)}{SD(y)} \cdot \beta_k$ (SD—standard deviation, $x_k$—$k$-th explanatory variable, $y$—explained variable), inter-

pretable as coefficients in Table 4 obtained after variable standardization. It allows for comparing the effects of one standard deviation increase on the dependent variable across regressors. One can conclude that, in quantitative terms, the Economic damage, followed by High blood pressure and R1b, contributed to the cross-regional variation in Cases to the greatest extent, whereas the contributions of Diabetes and Trust in health authorities remained most limited.

**Table 4.** Estimation results for the number of reported COVID-19 cases per 10,000 inhabitants.

|  | OLS | Mixed-W SARAR | Mixed-W SEM | SARAR | SEM |
|---|---|---|---|---|---|
| Economic damage | −9.546 *** | −9.588 *** | −11.264 *** | −9.406 *** | −10.751 *** |
|  | (1.329) | (2.523) | (2.412) | (1.897) | (1.813) |
| NACE B-E | 6.879 *** | 3.688 *** | 3.873 *** | 4.408 *** | 4.571 *** |
|  | (1.312) | (1.216) | (1.276) | (1.242) | (1.277) |
| High blood pressure | −9.727 *** | −7.652 *** | −7.598 *** | −7.997 *** | −8.210 *** |
|  | (1.810) | (2.758) | (2.917) | (2.153) | (2.314) |
| Chronic lower respiratory | −22.837 *** | −20.777 *** | −22.460 *** | −21.541 *** | −23.365 *** |
|  | (3.947) | (6.361) | (6.612) | (4.878) | (5.079) |
| Population aged 60 | −11.045 *** | −11.445 *** | −10.922 *** | −11.700 *** | −11.207 *** |
|  | (2.773) | (2.406) | (2.397) | (2.465) | (2.443) |
| Day | −1.310 *** | −0.643 ** | −0.658 ** | −0.777 *** | −0.809 *** |
|  | (0.310) | (0.263) | (0.265) | (0.253) | (0.252) |
| GDP | 1.127 *** | 0.733 *** | 0.779 *** | 0.761 *** | 0.807 *** |
|  | (0.301) | (0.255) | (0.255) | (0.261) | (0.259) |
| R1b | 1.666 ** | 2.828 ** | 3.906 *** | 2.457 ** | 3.125 *** |
|  | (0.662) | (1.351) | (1.436) | (0.971) | (1.022) |
| Trust in health authorities | −2.869 *** | −1.225 | −0.757 | −1.575 | −1.507 |
|  | (0.864) | (1.548) | (1.870) | (1.138) | (1.328) |
| Diabetes | 6.896 * | 3.512 | 1.650 | 2.866 | 1.728 |
|  | (4.013) | (6.501) | (7.055) | (5.143) | (5.637) |
| Constant | 1334.513 *** | 1153.576 *** | 1196.219 *** | 1227.974 *** | 1298.141 *** |
|  | (130.864) | (197.620) | (217.446) | (161.916) | (165.530) |
| $\rho_{borders}$ |  | 0.181 * |  | 0.149 |  |
|  |  | (0.096) |  | (0.092) |  |
| $\lambda_{borders}$ |  | 0.403 *** | 0.571 *** | 0.534 *** | 0.652 *** |
|  |  | (0.132) | (0.078) | (0.110) | (0.070) |
| $\lambda_{country}$ |  | 0.271 *** | 0.216 ** |  |  |
|  |  | (0.105) | (0.084) |  |  |
| Moran's I test |  |  |  |  |  |
| $W_{distance}$ | 6.725 *** | 0.483 | 0.274 | 0.837 | 0.688 |
| $W_{borders}$ | 7.315 *** | 0.376 | 0.266 | −0.135 | −0.191 |
| $W_{country}$ | 6.386 *** | 0.727 | 0.843 | 3.235 *** | 3.121 *** |

Notes: *** Significant at the 1 percent level. ** Significant at the 5 percent level. * Significant at the 10 percent level.

**Table 5.** Standardized regression coefficients for Cases.

|  | Economic Damage | NACE B-E | High Blood Pressure | Chronic Lower Respiratory | Population Aged 60 | Day | GDP | R1b | Trust in Health Authorities | Diabetes |
|---|---|---|---|---|---|---|---|---|---|---|
| mixed-W SARAR | −0.404 | 0.166 | −0.422 | −0.323 | −0.228 | −0.128 | 0.145 | 0.328 | −0.082 | 0.068 |
| mixed-W SEM | −0.474 | 0.174 | −0.419 | −0.349 | −0.218 | −0.131 | 0.155 | 0.454 | −0.050 | 0.032 |

Notes: Standardized regression coefficients, computed as $\frac{SD(x_k)}{SD(y)} \cdot \beta_k$. (SD—standard deviation, $x_k$—$k$-th explanatory variable, $y$—explained variable), $\beta_k$ as in Table 4.

### 3.2. Deaths Increase

Similarly to the number of cases, the percentage of deaths increase varied significantly across European countries and regions in the same country (see Figure 2). The smallest increase, which amounted to only around 1%, was recorded in Nordjylland, Denmark (DK05), while the largest was in Madrid, Spain (ES30), and equaled about 45%.

In Table 6, we presented the estimation results with three models. Column 2 summarizes the results of the general-to-specific approach. Contrary to the number of reported cases, a classic SEM model without another adjacency matrix was sufficient to capture each type of spatial autocorrelation. Nevertheless, in column 3, we presented the corresponding model with two neighboring matrices. Additionally, in column 1, we included the OLS estimates.

**Table 6.** Estimation results for the increase in the number of deaths.

|  | OLS | SEM | Mixed-W SEM |
|---|---|---|---|
| Cases | 0.026 *** | 0.026 *** | 0.026 *** |
|  | (0.002) | (0.003) | (0.003) |
| Pollution | 0.240 *** | 0.137 | 0.135 |
|  | (0.080) | (0.094) | (0.096) |
| Diabetes | 0.335 *** | 0.273 | 0.271 |
|  | (0.121) | (0.189) | (0.187) |
| Mean stringency | 0.275 *** | 0.229 ** | 0.228 ** |
|  | (0.060) | (0.094) | (0.093) |
| Day | −0.039 *** | −0.032 *** | −0.032 *** |
|  | (0.012) | (0.011) | (0.011) |
| Doctors | −0.007 ** | −0.005 ** | −0.005 ** |
|  | (0.003) | (0.003) | (0.003) |
| Health expenditure | −0.001 ** | −0.001 ** | −0.001 ** |
|  | (0.0005) | (0.001) | (0.001) |
| Constant | −12.079 ** | −6.690 | −6.428 |
|  | (5.622) | (8.788) | (8.917) |
| $\lambda_{borders}$ |  | 0.571 *** | 0.579 *** |
|  |  | (0.078) | (0.095) |
| $\lambda_{country}$ |  |  | −0.016 |
|  |  |  | (0.124) |
| Moran's I test |  |  |  |
| $W_{distance}$ | 4.045 *** | 0.384 | 0.389 |
| $W_{borders}$ | 5.717 *** | 0.863 | 0.839 |
| $W_{country}$ | 3.789 *** | 0.910 | 1.011 |

Notes: *** Significant at the 1 percent level. ** Significant at the 5 percent level.

Moran's I statistics for the simple linear model without spatial components indicate spatial autocorrelation related to each of the three analyzed matrices ($p < 0.0001$ for every matrix). However, including $W_{borders}$ is sufficient for capturing the whole autocorrelation of residuals ($p > 0.18$ for each of the three matrices in Moran's I test for the SEM model).

The coefficient estimate for the number of cases shows that each reported infection per 10,000 inhabitants was associated with an average increase in deaths in 2020 by 0.026 percentage points (with a 95% confidence interval equal [0.021, 0.031]).

Interestingly, the number of people with diabetes and average exposure to air pollution were significant in the non-spatial model but not in spatial ones. The lack of a correlation between air pollution and the number of COVID-19 fatalities is contrary to some ecological studies (Comunian et al. 2020; Prinz and Richter 2022). When air pollution is dropped from the analysis, no other variable emerges as additionally relevant, not least the population density.

Mean stringency is significant and positively correlated with deaths increase in all shown models. Probably, it is because the countries affected by the pandemic the most imposed stricter restrictions than the others. The impact of individual restrictions on

the number of cases and fatalities is hard to assess in a cross-sectional analysis like this. Although we tried the instrumental variable approach, described in detail in Appendix A.6, we recommend using panel analysis based on country-level data. Therefore, a far-fetched conclusion should not be drawn based on this positive correlation.

As can be concluded from the standardized regression coefficients in Table 7, the dominant contributor to explaining the cross-regional variation in the death count was the number of Cases, whereas the lowest contributions came from the number of Doctors, Pollution intensity, as well as diabetes incidence rate.

**Table 7.** Standardized regression coefficients for Deaths.

|  | Cases | Pollution | Diabetes | Mean Stringency | Day | Doctors | Health Expenditure |
|---|---|---|---|---|---|---|---|
| SEM | 0.593 | 0.107 | 0.123 | 0.187 | −0.146 | −0.093 | −0.215 |

*Notes:* Standardized regression coefficients, computed as $\frac{SD(x_k)}{SD(y)} \cdot \beta_k$ . (SD—standard deviation, $x_k$—$k$-th explanatory variable, $y$—explained variable), $\beta_k$ as in Table 6.

All the models for the deaths increase included Cases as one of the explanatory variables. Although this approach allows disaggregating the effect of COVID-19 on mortality into two separate channels, it does not allow for assessing the net effect of factors impacting both the number of cases and deaths per case. Therefore, we estimated different models for deaths increase without including Cases. The results are shown in Appendix A.8.

*3.3. Models Based on the Whole Set of Predictors*

A high correlation with other predictors could cause the significance of some variables in the models presented so far. To confirm that it did not happen, we estimated models with similar specifications to the previous ones but on the whole set of predictors. However, the number of regressors is quite large compared to the number of observations. Therefore, all the estimates presented in this subsection should be treated as complements of the previous two and not be interpreted separately.

We presented the estimation results in Table 8. To analyze infection incidence, we estimated mixed-W SEM and mixed-W SARAR as well as SEM for deaths increase. In the case of the number of reported cases, except R1b, all the significant variables in spatial models with preselected predictors are still significant at 10%. Furthermore, none of the variables rejected by BMA is significant in these settings.

An alternative way of explaining the impact of High blood pressure, Chronic lower respiratory disease, and Diabetes variables on the dependent ones could be based on the fact that the detectability of these diseases may depend on the healthcare system, i.e., when its quality is worse, the 'official' statistic underestimates the share of the chronically ill (better healthcare—more official cases of these three diseases). At the same time, it is straightforward to regard a better healthcare system dealing more efficiently with the outcomes of the COVID-19 pandemic (better healthcare—fewer deaths). Bottom line, when the three diseases in question act predominantly as a proxy for healthcare quality, a negative correlation with COVID-19 mortality arises. Nevertheless, this kind of risk should be predominantly related to the models explaining the number of deaths, and to a lesser extent—the number of cases.

On the other hand, in the case of the deaths increase model, only Cases, Day, Health expenditure, and Doctors are significant variables in both models based on the whole and the restricted sets of predictors.

The coefficients corresponding to the share of people working in the industry without construction were significantly greater than zero in each of the specifications concerning the number of cases (both with preselected and the whole sets of predictors). It could be caused by a greater chance of infection in crowded work establishments, or it was only a statistical effect stemming from mass testing campaigns in some workplaces such as mines in Silesia (Krzysztofik et al. 2020). However, if it was only a statistical effect, the larger share

of people employed in B-E sectors should inflate the number of deaths controlled by the number of cases, but this variable is insignificant in the model for the increase in deaths.

**Table 8.** Estimation results for the models based on the whole set of predictors.

| | Cases | | Deaths |
|---|---|---|---|
| | **Mixed-W SEM** | **Mixed-W SARAR** | **SEM** |
| Economic damage | −11.109 *** (2.523) | −9.488 *** (2.457) | −0.002 (0.090) |
| NACE B-E | 4.059 *** (1.360) | 4.234 *** (1.291) | 0.019 (0.060) |
| High blood pressure | −7.082 ** (3.112) | −8.052 *** (2.699) | −0.082 (0.107) |
| Chronic lower respiratory | −26.171 *** (8.727) | −25.076 *** (8.327) | 0.582 * (0.311) |
| Population aged 60 | −11.293 *** (2.570) | −11.979 *** (2.573) | −0.121 (0.125) |
| Day | −0.674 ** (0.274) | −0.717 *** (0.272) | −0.020 * (0.011) |
| GDP | 0.668 ** (0.306) | 0.552 * (0.308) | 0.009 (0.014) |
| R1b | 3.922 ** (1.880) | 1.864 (1.495) | 0.051 (0.052) |
| Trust in health authorities | −2.991 (2.747) | −2.579 (2.107) | 0.221 ** (0.088) |
| Diabetes | −1.789 (8.662) | 3.583 (6.561) | 0.506 * (0.261) |
| Cases | | | 0.023 *** (0.003) |
| Pollution | −1.225 (2.311) | −1.286 (2.283) | 0.231 ** (0.104) |
| Mean stringency | 1.907 (4.160) | 3.917 (3.677) | −0.021 (0.151) |
| Doctors | 0.023 (0.074) | 0.068 (0.077) | −0.009 *** (0.003) |
| Health expenditure | −0.004 (0.032) | 0.009 (0.025) | −0.002 ** (0.001) |
| Trust in government | 4.561 (4.691) | 2.244 (3.815) | −0.344 ** (0.142) |
| Density | 0.004 (0.009) | 0.004 (0.009) | 0.0004 (0.0004) |
| Constant | 1161.691 *** (308.130) | 985.806 *** (277.497) | 4.445 (11.455) |
| $\rho_{borders}$ | | 0.227 ** (0.104) | |
| $\lambda_{borders}$ | 0.597 *** (0.081) | 0.374 ** (0.147) | 0.599 *** (0.082) |
| $\lambda_{country}$ | 0.202 ** (0.086) | 0.256 ** (0.113) | |
| Moran's I test | | | |
| $W_{distance}$ | 0.326 | 0.718 | −0.232 |
| $W_{borders}$ | 0.043 | 0.332 | 0.776 |
| $W_{country}$ | 0.677 | 0.625 | −0.274 |

Notes: *** Significant at the 1 percent level. ** Significant at the 5 percent level. * Significant at the 10 percent level.

The number of days before the first cases negatively impacted the number of reported infections and the increase in the number of deaths in all models presented in the study. As far as the former, the correlation could be caused by the more extended period when people could infect. In the latter's case, this effect is controlled by the number of cases. Therefore, the negative impact of days on the deaths increase was probably caused by the fact that countries hit by COVID-19 later had more time to stock all the necessary resources used in the treatment and prepare a better pandemic strategy.

BMA selected neither health expenditure nor the number of doctors for the model explaining the number of cases, and none of these variables were significant in the models based on the whole set of predictors, which shows that their impact on the effective detection and prevention of infections was negligible. Nevertheless, both significantly reduced fatality rates for a given number of cases.

The coefficients for GDP per capita were significantly greater than zero in all models for the number of cases and insignificant in all models for deaths increase. This conclusion may be treated as an extension of the results obtained by Amdaoud et al. (2021) that showed that the level of GDP per capita was associated with the increased death rates but did not distinguish the effect between the impact on the infection rate and the case fatality rate.

## 4. Discussion

The study identifies and distinguishes factors that explain inter- and intra-national variation in COVID-19 mortality through two different mechanisms: increased infections and fatalities per case. Identifying them allows for the prediction of the outcome of the pandemic and the imposing of appropriate countermeasures by decision-makers. Countries and regions with better quality healthcare, measured by the number of practitioners and

health expenditures, could keep death rates low with a relatively high number of cases. On the other hand, entities with less developed healthcare systems could only try to decrease the number of infections. However, when imposing restrictions to reduce the number of cases, the decision-makers have to account for specific characteristics of societies that concern the population structure, employment structure, wealth, or attitudes toward these limitations.

It must be emphasized that the public health implications of this study do not generalize easily to other periods or geographic areas. From 2021 onward, the landscape has changed considerably with the increasing availability of vaccines, virus mutations (mostly towards endemization), and increasing organizational preparedness of the healthcare systems. Probably, a comparable setting in which these results apply would be an emergence of a new respiratory pathogen (or a milestone mutation of an existing one) with at least similar properties in terms of transmission and severity. Peters (2022) summarizes the evidence that such a scenario is non-negligible.

Our estimation results confirm that accounting for spatial autocorrelation is necessary for analyzing the outcomes of the pandemic, which is in line with Amdaoud et al. (2021). What is methodologically new in our study is the use of two spatial adjacency matrices simultaneously. An additional component indicating adherence to the same country is introduced mainly to enable the inclusion of variables from the higher level of aggregation. However, even with all attributes at the same level of aggregation employing this matrix may be helpful due to the possible existence of some unobserved and country-specific factors that impact the outcome of the pandemic.

The analysis in the study can be extended in several ways. Firstly, new variants of the SARS-CoV-2 may respond differently to factors that were important in the first waves of the pandemic. Therefore, further studies can include different periods. However, in this approach analyzing panel data may be more appropriate because the pace of allocation of vaccines, which is a critical factor in predicting the outcome of the pandemic from the beginning of 2021, cannot be easily aggregated with the cross-sectional data.

Secondly, as shown in the study, employing panel econometrics may also be helpful because different countries imposed various restrictions at different moments, and aggregating them into simple, cross-sectional indices seems insufficient.

Finally, all the neighboring matrices analyzed in the study are based on the geographical proximity of regions or adherence to the same country. However, as suggested by Kandula and Shaman (2021), using matrices that reflect the flows of people between regions can also be considered. Such matrices, however, are difficult to obtain in practice (see Torój (2022), Table 7 for a scarce example for NUTS-3 regions in Poland), at least not the entire Europe as in this study. One might also expect them to resemble the distance-based matrix, explored here in Appendix A.5.

**Author Contributions:** The authors contributed to the study's conception and design. Material preparation, data collection, and analysis were performed by M.S. The first draft of the manuscript was written by M.S., and the authors all commented on previous versions. The authors read and approved the final manuscript. The study was supervised by A.T. All authors have read and agreed to the published version of the manuscript.

**Funding:** This research received no external funding.

**Data Availability Statement:** All the data and scripts for estimating the models are available here: https://github.com/matszsz/COVID_spatial (accessed on 14 June 2023).

**Conflicts of Interest:** The authors declare no conflicts of interest.

## Appendix A. Additional Materials

*Appendix A.1. Construction of Selected Variables and Missing Data Imputation*

The measurements for the Doctors variable were based on the following years: 2019 (Germany, France, Czech Republic, Spain, Belgium, Austria, Netherlands, Portugal, Greece, Italy, Romania, Hungary, Latvia, and Slovakia), 2018 (Denmark, Estonia, Poland without the PL91 and the PL92 regions, and Sweden) and 2014 (Finland). Since the PL91 and the PL92 regions were first introduced in the 2018 version of NUTS coding, there were no available measurements for them in the dataset. Therefore, we assumed that for both regions, it equaled 281.13, which is the most recent value (from 2016) for the PL12 region from the 2015 NUTS coding (before 2018, the PL91 and the PL92 were parts of the PL12).

In 2020, Kantar conducted surveys on public opinion in the EU in times of coronavirus crisis. The first edition took place in April and covered 21 countries, and the second and third editions in July and October and covered 27 countries. They aimed to identify differences between nations in their attitudes towards actions taken by governments and the European Union concerning the pandemic. From multiple questions, we selected a few that were used for creating behavioral variables. Due to the larger number of covered countries, we based the analysis on the second and the third editions.

One of the questions was as follows: "Where do you position yourself between these two statements regarding the consequences of the restriction measures in your country?". Respondents were supposed to choose the response from the 6-point Likert Scale, where 1 corresponded to "The health benefits are greater than the economic damage" and 6 to "The economic damage is greater than the health benefits". We summed the share of people from the third edition of the survey that chose 1 or 2, and in that way, we created the economic damage variable. Other approaches to aggregation were possible, but they did not change the final results (see Appendix A.4).

Another question was as follows "From the following list, who do you trust most to inform you about the coronavirus pandemic?". The respondents could choose up to three responses from: "Scientists", "The country's health authorities", "World Health Organization", "The country's government", "My doctor", "My family members and friends", "Journalists from traditional media such as newspapers, radio, TV", "Non-governmental organizations (NGOs) working on health and social issues", "EU institutions such as the European Commission or the European Parliament", "Local and regional authorities", "My pharmacist", "Citizens for example on online social networks", or "Other". We checked the shares of people in the second edition that chose "The country's health authorities" and "The country's government". Based on that, we created trust in health authorities and trust in government variables. This question was not asked in the third edition of the study.

*Appendix A.2. Scatterplots*

To check if the significance of parameter estimates in the final models is not caused by the correlation with other variables, in Figures A1–A3 we showed scatterplots with each of the explanatory variables and the dependent ones. Additionally, we added regression lines determined by the OLS.

It is especially worth noticing that the relation between the Population aged 60 and the number of cases is strongly negative, whereas the association between both variables concerning the reported health condition (Chronic lower respiratory disease and High blood pressure) and Cases is moderately negative. These results are in line with the conclusions from the final models for the reported number of cases in which all of these variables negatively correlated with the dependent variable.

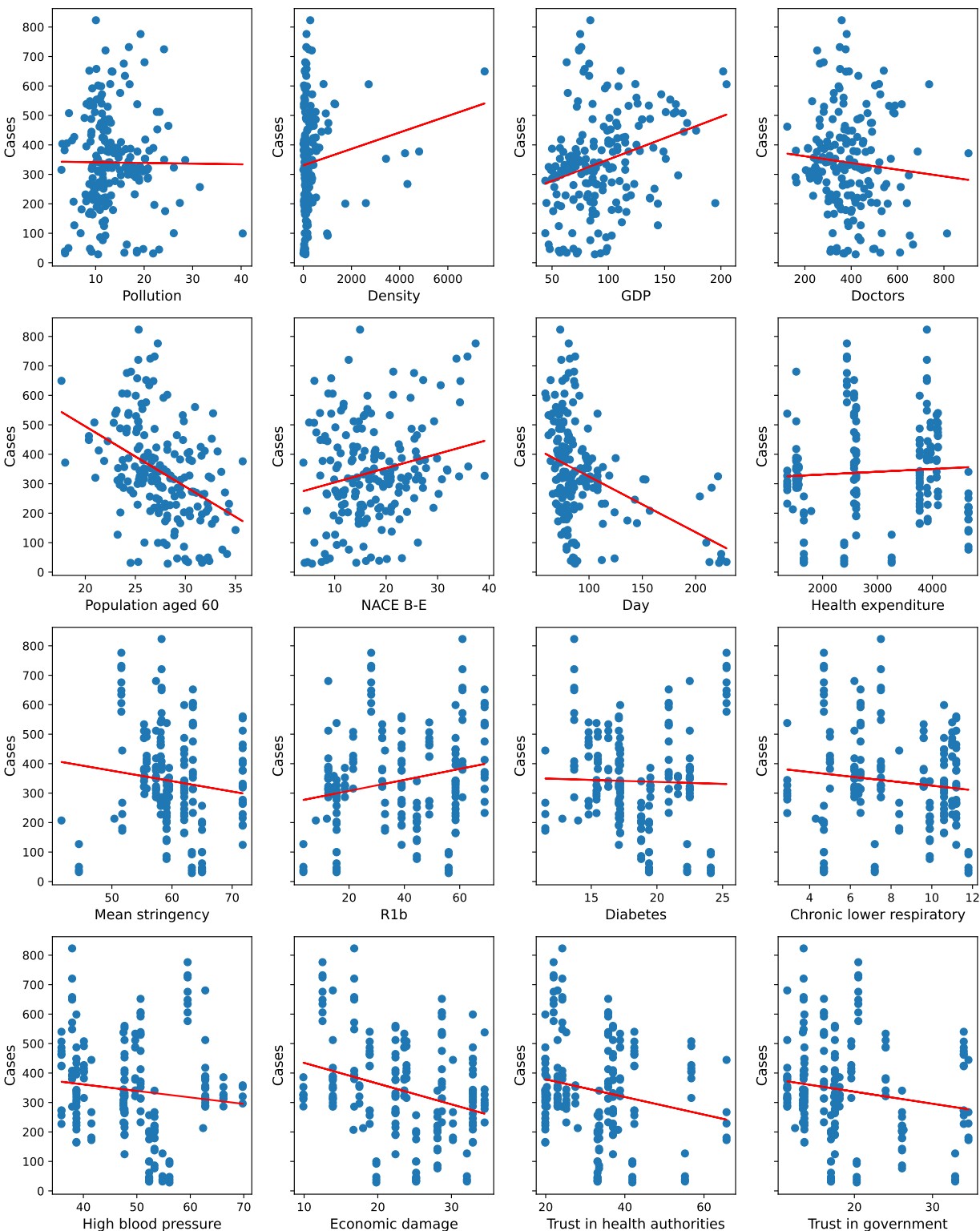

**Figure A1.** Number of reported COVID-19 cases per 10,000 inhabitants in 2020 in analyzed EU regions against different explanatory variables.

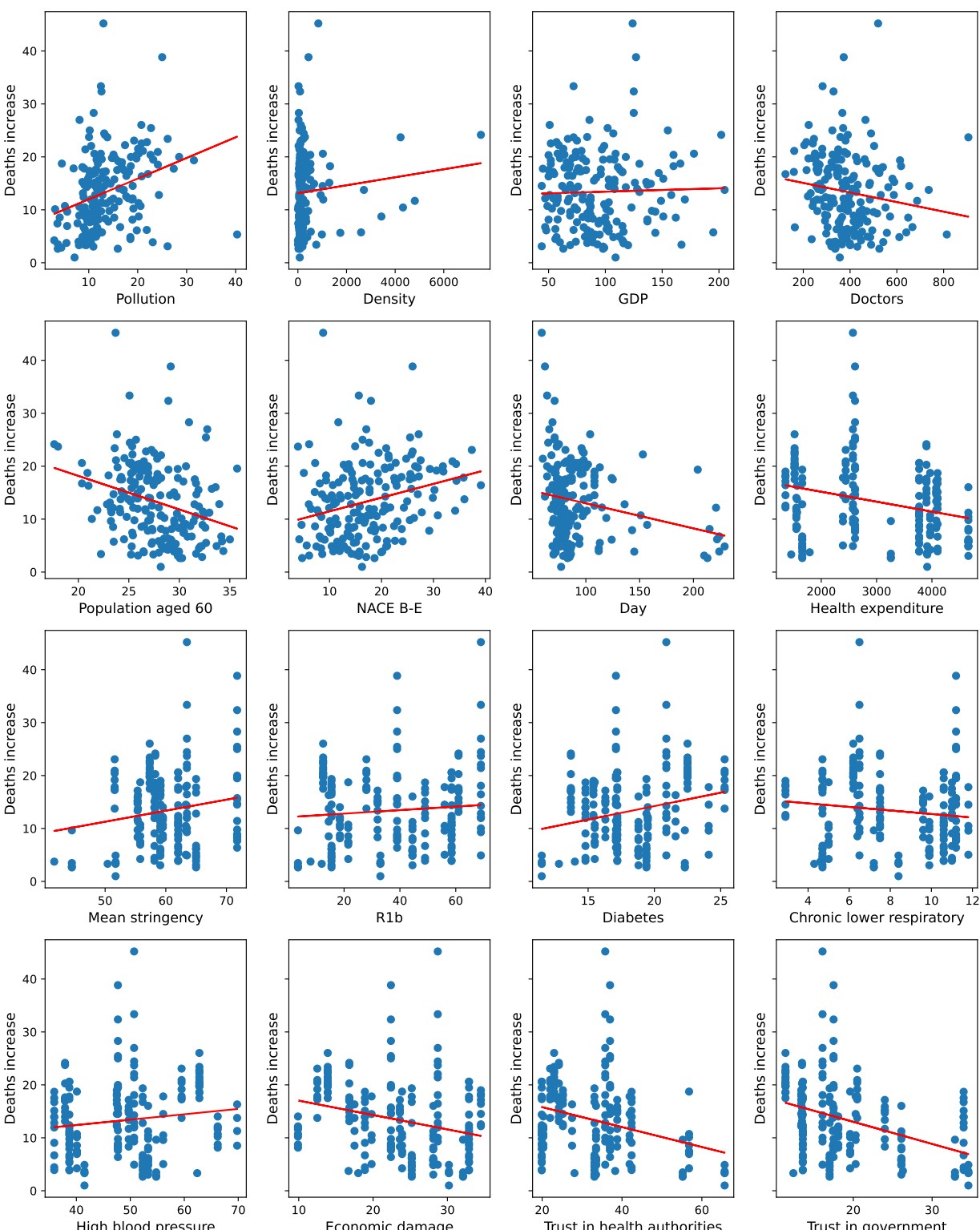

**Figure A2.** Percentage increase in the number of deaths in analyzed EU regions in 2020 compared with the average number of deaths between 2016 and 2019 against different explanatory variables.

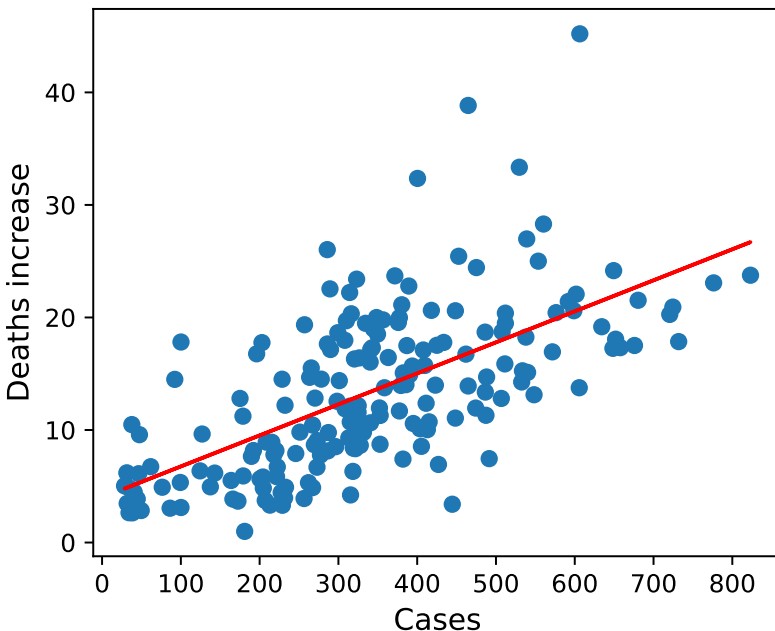

**Figure A3.** Percentage increase in the number of deaths in analyzed EU regions in 2020 compared with the average number of deaths between 2016 and 2019 against the reported number of infections.

*Appendix A.3. Variance Inflation Factor*

We calculated the variance inflation factor (VIF) for both sets of OLS estimates: explaining the relative number of cases and the deaths increase. In the former, the values of VIF for Economic damage, NACE B-E, Chronic lower respiratory disease, Population aged 60, Day, GDP, and Trust in health authorities were smaller than 2. For High blood pressure, it equaled 4.68, for R1b 2.79, and for Diabetes 2.87. In the latter, the values of VIF for Cases, Pollution, Diabetes, Mean stringency, Day, and Doctors were smaller than 2, while in the case of Health expenditure, it equaled 2.2. For each variable, the value of VIF was smaller than 5. Therefore, we concluded that the multicollinearity problem did not affect the results importantly.

*Appendix A.4. Alternative Versions of Constructing Economic Damage Variable*

We estimated the final models for the number of cases based on other economic damage variable aggregation methods. The summary was included in Table A1. In columns 1 and 2, we presented mixed-W SARAR and mixed-W SEM based on the variable created from the second edition of the survey (instead of the third one as in the baseline approach). In columns 3 and 4, we presented models based on the third edition but with the share of 1-3 options instead of 1-2. Regardless of the aggregation method, the variable coefficient is significantly smaller than zero (with $p < 0.001$ in each case).

**Table A1.** Estimation results for number of cases with different aggregation approaches to economic damage variable.

| | 2nd Edition (1-2) | | 3rd Edition (1-3) | |
|---|---|---|---|---|
| | **Mixed-W SARAR** | **Mixed-W SEM** | **Mixed-W SARAR** | **Mixed-W SEM** |
| Economic damage | −6.797 *** | −7.947 *** | −7.567 *** | −8.719 *** |
| | (1.961) | (1.930) | (1.880) | (1.809) |
| NACE B-E | 3.583 *** | 3.933 *** | 3.658 *** | 3.874 *** |
| | (1.207) | (1.265) | (1.212) | (1.267) |
| High blood pressure | −7.357 ** | −7.958 *** | −8.431 *** | −8.296 *** |
| | (2.876) | (3.059) | (2.762) | (2.923) |
| Chronic lower respiratory | −22.070 *** | −24.598 *** | −19.483 *** | −20.710 *** |
| | (6.734) | (6.943) | (6.179) | (6.539) |
| Population aged 60 | −11.976 *** | −11.531 *** | −11.469 *** | −10.972 *** |
| | (2.413) | (2.435) | (2.402) | (2.396) |
| Day | −0.595 ** | −0.630 ** | −0.643 ** | −0.663 ** |
| | (0.264) | (0.263) | (0.262) | (0.264) |
| GDP | 0.701 *** | 0.753 *** | 0.732 *** | 0.787 *** |
| | (0.256) | (0.258) | (0.255) | (0.254) |
| R1b | 2.629 ** | 3.510 ** | 2.588 ** | 3.613 *** |
| | (1.275) | (1.387) | (1.298) | (1.398) |
| Trust in health authorities | −1.420 | −0.942 | −1.484 | −1.041 |
| | (1.563) | (1.933) | (1.512) | (1.843) |
| Diabetes | 4.849 | 3.638 | 3.866 | 2.140 |
| | (6.514) | (7.303) | (6.395) | (6.974) |
| Constant | 1127.565 *** | 1208.533 *** | 1298.653 *** | 1346.529 *** |
| | (207.619) | (225.968) | (208.594) | (227.147) |
| $\rho_{borders}$ | 0.212 ** | | 0.183 * | |
| | (0.093) | | (0.095) | |
| $\lambda_{borders}$ | 0.348 *** | 0.545 *** | 0.402 *** | 0.571 *** |
| | (0.131) | (0.080) | (0.130) | (0.078) |
| $\lambda_{country}$ | 0.321 *** | 0.247 *** | 0.264 ** | 0.210 ** |
| | (0.103) | (0.081) | (0.105) | (0.085) |
| Moran's I test | | | | |
| $W_{distance}$ | 0.369 | 0.309 | 0.373 | 0.139 |
| $W_{borders}$ | 0.547 | 0.480 | 0.372 | 0.297 |
| $W_{country}$ | 0.701 | 0.904 | 0.733 | 0.905 |

Notes: *** Significant at the 1 percent level. ** Significant at the 5 percent level. * Significant at the 10 percent level.

*Appendix A.5. Models with Neighboring Matrices Based on Distance*

In Table A2, we showed mixed-W SARAR models for the number of cases, and deaths increase with $W_{distance}$ instead of $W_{borders}$. In the case of the former, $\rho_{distance}$ and $\lambda_{distance}$ are significant at 1%. However, the spatial autocorrelation based on $W_{distance}$ and $W_{borders}$ is not captured. In the latter, $\rho_{distance}$ and $\lambda_{distance}$ are insignificant, and the spatial autocorrelation is also not captured. Therefore, we argue that using $W_{borders}$ is more appropriate for analyzing both dependent variables.

**Table A2.** Estimation results for the models with $W_{distance}$ instead of $W_{borders}$.

|  | **Cases** | **Deaths** |
|---|---|---|
| Economic damage | −9.789 *** (2.120) |  |
| NACE B-E | 4.293 *** (1.222) |  |
| High blood pressure | −10.060 *** (2.631) |  |
| Chronic lower respiratory | −23.653 *** (5.875) |  |
| Population aged 60 | −10.854 *** (2.451) |  |
| Day | −0.859 *** (0.292) | −0.026 * (0.013) |
| GDP | 0.802 *** (0.269) |  |
| R1b | 2.240 ** (1.089) |  |
| Trust in health authorities | −1.502 (1.276) |  |
| Diabetes | 7.610 (5.913) | 0.317 * (0.172) |
| Cases |  | 0.027 *** (0.003) |
| Pollution |  | 0.266 *** (0.092) |
| Mean stringency |  | 0.262 *** (0.087) |
| Doctors |  | −0.006 ** (0.003) |
| Health expenditure |  | −0.001 (0.001) |
| Constant | 989.046 *** (227.834) | −19.745 ** (7.965) |
| $\rho_{distance}$ | 0.733 *** (0.210) | 0.436 (0.347) |
| $\lambda_{distance}$ | 0.556 *** (0.164) | 0.436 (0.306) |
| $\lambda_{country}$ | 0.374 *** (0.110) | 0.293 * (0.151) |
| Moran's I test |  |  |
| $W_{distance}$ | 3.475 *** | 2.220 ** |
| $W_{borders}$ | 5.158 *** | 4.171 *** |
| $W_{country}$ | 0.719 | 0.677 |

Notes: *** Significant at the 1 percent level. ** Significant at the 5 percent level. * Significant at the 10 percent level.

*Appendix A.6. Instrumental Variable for Mean Stringency*

Due to the fact the COVID-19 restrictions policy in many countries is based on the previous, current, and predicted number of infections and deaths, the Mean stringency variable might be endogenous, which could impact the estimates of parameters in the models. Therefore we introduced the instrumental variable that is based on the following model:

$$\hat{y}_{t,i} = \beta_1 \cdot y_{t-7,-i} + c^T \cdot \beta_2, \tag{A1}$$

where $\hat{y}_{t,i}$ is the stringency of restrictions in the country $i$ in day $t$, $y_{t-7,-i}$, is the average stringency of restrictions 7 days earlier in all analyzed countries except $i$, and $c^T$ is a vector of 19 binary variables whose elements indicate different countries. This approach relies on the assumption proposed by Fakir and Bharati (2021) that restrictions in other countries impact the reported number of infections and deaths increase only via adjusting restrictions in our country.

We estimated this model with the OLS estimator (obtained $R^2$ is equal to around 0.96) and predicted the number of restrictions for each country and each day from the beginning of the pandemic till the end of the year. Then, we calculated the average predicted mean stringency, which is the instrumental variable for the Mean stringency variable. The following steps were equivalent to the ones shown in the main body of the text but with the instrumental variable instead of the initial one. The summary of BMS is shown in Tables A3 and A4.

**Table A3.** Bayesian Model Averaging for the number of infections per 10,000 people with instrumental variable.

|  | PIP | Post Mean | Post SD | Cond.Pos.Sign |
| --- | --- | --- | --- | --- |
| Economic damage | 1.000 | −9.580 | 1.338 | 0.000 |
| NACE B-E | 1.000 | 7.022 | 1.399 | 1.000 |
| High blood pressure | 1.000 | −8.826 | 2.208 | 0.000 |
| Chronic lower respiratory disease | 1.000 | −22.264 | 4.785 | 0.000 |
| Population aged 60 | 0.997 | −11.731 | 2.960 | 0.000 |
| Day | 0.995 | −1.304 | 0.360 | 0.000 |
| GDP | 0.986 | 1.207 | 0.362 | 1.000 |
| R1b | 0.834 | 1.598 | 0.962 | 1.000 |
| Trust in health authorities | 0.768 | −2.232 | 1.536 | 0.000 |
| Diabetes | 0.464 | 3.717 | 4.901 | 1.000 |
| Trust in government | 0.375 | −1.276 | 2.123 | 0.011 |
| Mean stringency (IV) | 0.361 | 1.174 | 1.979 | 1.000 |
| Density | 0.269 | −0.004 | 0.010 | 0.002 |
| Doctors | 0.170 | 0.012 | 0.046 | 0.988 |
| Health expenditure | 0.158 | −0.001 | 0.007 | 0.294 |
| Pollution | 0.148 | −0.072 | 0.965 | 0.467 |

**Table A4.** Bayesian Model Averaging for the death toll increase with instrumental variable.

|  | PIP | Post Mean | Post SD | Cond.Pos.Sign |
| --- | --- | --- | --- | --- |
| Cases | 1.000 | 0.027 | 0.003 | 1.000 |
| Mean stringency (IV) | 0.864 | 0.257 | 0.130 | 1.000 |
| Pollution | 0.826 | 0.235 | 0.140 | 1.000 |
| Diabetes | 0.820 | 0.345 | 0.202 | 1.000 |
| Doctors | 0.620 | −0.005 | 0.005 | 0.000 |
| Day | 0.568 | −0.019 | 0.020 | 0.000 |
| Health expenditure | 0.447 | −0.001 | 0.001 | 0.000 |
| Trust in government | 0.306 | −0.067 | 0.124 | 0.000 |
| Trust in health authorities | 0.234 | 0.028 | 0.062 | 0.957 |
| Population aged 60 | 0.227 | −0.053 | 0.117 | 0.001 |
| Chronic lower respiratory disease | 0.183 | 0.068 | 0.206 | 0.825 |
| Economic damage | 0.113 | 0.009 | 0.032 | 0.992 |
| Density | 0.111 | 0.000 | 0.000 | 0.968 |
| R1b | 0.099 | 0.003 | 0.015 | 0.799 |
| GDP | 0.099 | 0.002 | 0.007 | 0.914 |
| NACE B-E | 0.067 | 0.000 | 0.017 | 0.459 |

Replacing Mean stringency with its instrument did not affect the results importantly. In the case of the model explaining the number of reported cases, Mean stringency (IV) was below the threshold of 0.4 and should not be included in the final models. On the other hand, in the model for deaths increase, it was above the threshold and did not change the subset of chosen variables. Therefore, we estimated an analogous SEM model to the one from Table 6 and showed it in Table A5.

Finally, the instrumental variable did not change the signs or the significance of the parameters in the models. The lack of the positive impact of imposing restrictions in the results could be caused by the fact the stringency index combines multiple factors that impact the transmission of the virus differently. Moreover, apart from the level of stringency, the timing of imposing restrictions is essential. Therefore, we can conclude that assessing the impact of restrictions on the number of cases and deaths should be based on the panel data with the country-level aggregation rather than the cross-sectional data on the regional level. However, it is not the topic of this analysis.

**Table A5.** Estimation results for the increase in the number of deaths (with the instrumental variable).

|  | SEM |
|---|---|
| Cases | 0.026 *** |
|  | (0.003) |
| Pollution | 0.138 |
|  | (0.094) |
| Diabetes | 0.281 |
|  | (0.188) |
| Mean stringency (IV) | 0.247 ** |
|  | (0.096) |
| Day | −0.031 *** |
|  | (0.011) |
| Doctors | −0.006 ** |
|  | (0.003) |
| Health expenditure | −0.001 ** |
|  | (0.001) |
| Constant | −8.025 |
|  | (8.916) |
| $\lambda_{borders}$ | 0.568 *** |
|  | (0.078) |
| Moran's I test |  |
| $W_{distance}$ | 0.381 |
| $W_{borders}$ | 0.863 |
| $W_{country}$ | 0.863 |

Notes: *** Significant at the 1 percent level. ** Significant at the 5 percent level.

*Appendix A.7. SLX Specification*

To further check the appropriateness of choosing mixed-W SARAR as the primary model in the analysis, we compared it with the mixed-W SLX specification, whose parameters were estimated with the OLS estimator. Therefore we added spatial lags to the linear model for each explanatory variable. The predictors aggregated at the regional level lagged with respect to both matrices, whereas the variables at the country level were lagged with respect to the matrix based on the queen congruity. In this case, we decided not to include the other matrix because the spatial lags for these variables would be equivalent to the corresponding unlagged variables (except for isolated regions, for which it would always equal 0).

The summary of both SLX models for the number of cases and the deaths increase is shown in Tables A6 and A7. As can be seen from the results of Moran's I test, none of the models could account for the spatial autocorrelation with respect to any of the analyzed matrices. What is more, the values of the Akaike information criterion (AIC) are seemingly higher for SLX models (for the number of cases, it equals 2299.09, while it was 2240 for mixed-W SARAR and 2243.36 for mixed-W SEM, for the deaths, increase it equals 1127.67 while it was 1080.11 for SEM) which indicates that they fit much worse to the data. Finally, most of the parameter estimates are not significant, and many of the ones that are do not allow for any meaningful interpretation (like the fact that the level of pollution impacts the deaths increase in neighboring regions, but not on its own). Therefore, we can conclude that the mixed-W SARAR specification shown in the main body of the paper describes the analyzed problem much better than the SLX model.

**Table A6.** Estimation results for the number of reported COVID-19 cases per 10,000 inhabitants with SLX specification.

|  | SLX |
|---|---|
| Economic damage | −10.839 *** (2.887) |
| NACE B-E | 3.438 ** (1.735) |
| High blood pressure | −10.929 *** (3.104) |
| Chronic lower respiratory disease | −26.811 *** (8.112) |
| Population aged 60 | −11.871 *** (2.982) |
| Day | −0.769 * (0.400) |
| GDP | 0.363 (0.343) |
| R1b | 5.952 *** (1.743) |
| Trust in health authorities | 3.756 (2.535) |
| Diabetes | −4.820 (9.351) |
| lag.borders NACE B-E | 3.179 (3.415) |
| lag.borders Population aged 60 | 3.391 (5.495) |
| lag.borders Day | −1.036 (0.713) |
| lag.borders GDP | 2.054 *** (0.722) |
| lag.countries NACE B-E | 10.936 ** (4.345) |
| lag.countries Population aged 60 | −1.555 (6.481) |
| lag.countries Day | 0.499 (0.907) |
| lag.countries GDP | −1.535 ** (0.733) |
| lag.borders Economic damage | 1.437 (3.526) |
| lag.borders High blood pressure | −2.973 (4.027) |
| lag.borders Chronic lower respiratory disease | 3.356 (9.433) |
| lag.borders R1b | −4.384 ** (1.764) |
| lag.borders Trust in health authorities | −5.491 ** (2.464) |
| lag.borders Diabetes | 13.868 (11.073) |
| Constant | 1281.337 *** (159.646) |
| Moran's I test | |
| $W_{distance}$ | 4.494 *** |
| $W_{borders}$ | 6.732 *** |
| $W_{country}$ | 3.268 *** |

Notes: *** Significant at the 1 percent level. ** Significant at the 5 percent level. * Significant at the 10 percent level.

**Table A7.** Estimation results for the increase in the number of deaths with SLX specification.

|  | SLX |
|---|---|
| Cases | 0.029 *** (0.004) |
| Pollution | 0.146 (0.129) |
| Diabetes | 0.290 (0.296) |
| Mean stringency | 0.340 *** (0.116) |
| Day | −0.031 * (0.017) |
| Doctors | −0.007 ** (0.003) |
| Health expenditure | −0.002 ** (0.001) |
| lag.borders Cases | 0.003 (0.005) |
| lag.borders Pollution | 0.435 ** (0.194) |
| lag.borders Day | −0.016 (0.029) |
| lag.borders Doctors | −0.011 (0.007) |
| lag.countries Cases | −0.010 ** (0.004) |
| lag.countries Pollution | −0.280 (0.210) |
| lag.countries Day | −0.016 (0.029) |
| lag.countries Doctors | 0.009 (0.007) |
| lag.borders Diabetes | 0.036 (0.307) |
| lag.borders Mean stringency | −0.092 (0.111) |
| lag.borders Health expenditure | 0.001 (0.001) |
| Constant | −7.344 (6.791) |
| Moran's I test | |
| $W_{distance}$ | 2.764 *** |
| $W_{borders}$ | 4.806 *** |
| $W_{country}$ | 3.572 *** |

Notes: *** Significant at the 1 percent level. ** Significant at the 5 percent level. * Significant at the 10 percent level.

*Appendix A.8. Net Effects of Severity and Incidence*

The analyzed variables could impact COVID-19 mortality directly or via increased incidence rate. Moreover, the direction of these effects could be opposite, e.g., some variables could reduce the number of cases but increase the per-case mortality. Therefore, it is interesting to see what the net effect of each of the factors is. To do so, we estimated the model explaining the deaths increase but contrary to the previous settings, omitted the Cases variable.

As previously performed, we used the BMA to detect features with posterior inclusion probability above 0.4 and used them to estimate the models. Table A8 shows the results of the feature selection, and Table A9 the estimates of the models. The variables R1b, NACE B-E, and Doctors are insignificant in the spatial models, while the Trust in government negatively correlates with the deaths increase (contrary to the version of the model including Cases, the variable was selected by BMA). Interestingly, the Pollution variable is significant in all the presented settings. Additionally, it is worth noticing that in the mixed-W SARAR model, $\lambda_{country}$ is insignificantly different from zero. However, when excluding $W_{country}$ from the model, Moran's I test suggests that the model has remaining autocorrelation stemming from adherence to the same country.

**Table A8.** Bayesian Model Averaging for the death toll increase with the Cases variable excluded.

|  | PIP | Post Mean | Post SD | Cond.Pos.Sign |
|---|---|---|---|---|
| Day | 0.998 | −0.077 | 0.019 | 0.000 |
| Pollution | 0.951 | 0.360 | 0.137 | 1.000 |
| Population aged 60 | 0.934 | −0.463 | 0.199 | 0.000 |
| Trust in government | 0.914 | −0.291 | 0.132 | 0.000 |
| R1b | 0.532 | 0.045 | 0.050 | 1.000 |
| NACE B-E | 0.479 | 0.074 | 0.090 | 1.000 |
| Doctors | 0.425 | −0.004 | 0.006 | 0.000 |
| Economic damage | 0.383 | −0.054 | 0.081 | 0.000 |
| Mean stringency | 0.361 | 0.089 | 0.147 | 0.995 |
| GDP | 0.330 | 0.013 | 0.023 | 1.000 |
| Health expenditure | 0.315 | −0.0005 | 0.001 | 0.004 |
| Diabetes | 0.270 | 0.080 | 0.160 | 1.000 |
| Trust in health authorities | 0.174 | 0.013 | 0.048 | 0.833 |
| Chronic lower respiratory | 0.150 | −0.048 | 0.185 | 0.165 |
| Density | 0.093 | −0.00002 | 0.0002 | 0.161 |

**Table A9.** Estimation results for the increase in the number of deaths with the Cases excluded.

|  | OLS | Mixed-W SARAR | SARAR |
|---|---|---|---|
| Day | −0.074 *** | −0.037 *** | −0.039 *** |
|  | (0.015) | (0.013) | (0.013) |
| Pollution | 0.453 *** | 0.210 ** | 0.212 ** |
|  | (0.093) | (0.106) | (0.105) |
| Population aged 60 | −0.521 *** | −0.547 *** | −0.544 *** |
|  | (0.126) | (0.119) | (0.118) |
| Trust in government | −0.303 *** | −0.206 ** | −0.212 ** |
|  | (0.070) | (0.091) | (0.084) |
| R1b | 0.068 *** | 0.055 | 0.051 |
|  | (0.026) | (0.037) | (0.035) |
| NACE B-E | 0.141 ** | 0.098 | 0.096 |
|  | (0.063) | (0.064) | (0.064) |
| Doctors | −0.005 | −0.003 | −0.003 |
|  | (0.004) | (0.003) | (0.003) |
| Constant | 31.068 *** | 26.885 *** | 27.401 *** |
|  | (4.732) | (5.348) | (5.168) |

**Table A9.** *Cont.*

| | OLS | Mixed-W SARAR | SARAR |
|---|---|---|---|
| $\rho_{borders}$ | | 0.278 *** | 0.270 *** |
| | | (0.105) | (0.105) |
| $\lambda_{borders}$ | | 0.462 *** | 0.496 *** |
| | | (0.131) | (0.117) |
| $\lambda_{country}$ | | 0.076 | |
| | | (0.122) | |
| Moran's I test | | | |
| $W_{distance}$ | 3.762 *** | 0.218 | 0.252 |
| $W_{borders}$ | 6.701 *** | 0.610 | 0.312 |
| $W_{country}$ | 3.835 *** | 0.917 | 1.467 * |

Notes: *** Significant at the 1 percent level. ** Significant at the 5 percent level. * Significant at the 10 percent level.

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
