# Peer review of "Socio-Economic and Demographic Factors Associated with COVID-19 Mortality in European Regions: Spatial Econometric Analysis"

_econometrics, doi:10.3390/econometrics11020017_

Round 1

Reviewer 1 Report

I have the reviewed econometrics-2280610 and have the following major and minor comments for the authors:

* Major comments

1. A regression model for deaths without cases included would have a different interpretation, estimating the total rather than the direct effect of the different factors. As it is now, one cannot tell from table 6 whether regions with more vulnerable population groups were more severely hit in terms of excess mortality. I therefore suggest that estimates of the total effect are added, at least as an appendix.

2. The model for estimating expected mortality based on the 2016-2019 average is crude. As a sensitivity analysis and robustness check, I suggest a linear trend is used (see e.g. Kepp et al. Int Jrn Epi for a lengthy discussion of how sensitive excess mortality estimations can be to choice of model).

3. I lack a discussion of effect sizes – the paper focuses only on statistical significance. Effect sizes could for example be discussed in relation to the typical variation (say IQR) in a particular factor seen across the regions.

4. The positive association between trust in health authorities and deaths in Table 6 is unexpected. On the other hand, the association with the number of cases is negative, so again the net (total) effect would have been interesting to see.

5. To what extent does the positive association between air pollution and deaths reflect population density? Does the association with density change markedly if air pollution is omitted from the models?

6. Discussion row 407: It is claimed that countries and regions with better healthcare quality could keep death rates low with a relative high number of cases. This is a conclusion that does not follow naturally from the presented results but is something that could be investigated in an interaction analysis, for example by regressing deaths stratified on case load (low/medium/high).

* Minor comments

1. General - explain units of the independent and dependent variables in all tables and figures, or refer the reader to table 1 if space do not permit as in Figure A1 and A2. 

2. Row 42 - Björk et al. 2021 is not correctly cited (this paper is about regional variation in the winter holiday week in relation to excess mortality)

3. Row 81 - I would suggest moving paragraph on environmental factors (air pollution) as a risk factor to row 61 such that it comes before the paragraphs on variation in policy and restrictions.

4. Row 179 - explain the abbreviation NACE B-E first time it is mentioned in the text. 

Author Response

Dear Editors and Reviewers,

we are grateful for all the insightful comments - they allowed us to spot room for improvement in multiple dimensions, learn different perspectives, and generally raise the quality of the manuscript as compared to the pre-revision version. Please find below the detailed responses to the comments and the summary of our reactions to these comments. Needless to say, we are happy to introduce further changes if needed.

Authors

Reviewer 2 Report

The authors provide a comprehensive introduction to the topic and review relevant literature on various factors affecting morbidity and mortality. The proposed research question is highly relevant, given the recent COVID-19 pandemic.

The introduction provides a clear overview of the background and context of the study.

The literature review is extensive, covering various aspects of the COVID-19 pandemic and its effects on morbidity and mortality.

The authors justify their choice of the SARAR model and its suitability for the analysis of COVID-19 cases and deaths in European regions.

The introduction could benefit from a more explicit statement of the main research question and the specific objectives of the study.

While the authors mention the potential for spurious spatial autocorrelation due to using variables from different aggregation levels, they do not provide a solution to this issue. It would be valuable to discuss potential methods to address this problem or to justify the use of variables from different levels of aggregation despite this limitation.

The authors discuss the potential impact of variables like High blood pressure, Chronic lower respiratory disease, and Diabetes on the dependent variables, and suggest that the detectability of these diseases may depend on the healthcare system. However, the interpretation of these variables and their relationship with the healthcare system is not entirely clear, and it could benefit from further clarification.

The authors mention correlations between variables, but it is important to note that correlation does not necessarily imply causation. Establishing causal relationships would require additional analysis, such as using instrumental variables or other techniques to address endogeneity concerns.

While the authors present results based on different models and specifications, it would be helpful to conduct further robustness checks to ensure that the findings are consistent across various model specifications and estimation techniques. This could increase confidence in the study's conclusions.

The study focuses on specific countries and a particular period during the COVID-19 pandemic, and the authors acknowledge that the findings may not hold for different periods or new variants of the virus. They suggest extending the analysis to include different periods and using panel data, but it is crucial to discuss the limitations of the current study in terms of generalizability to other settings and time periods.

The authors mention the possibility of using matrices that reflect the flows of people between regions, as suggested by Kandula and Shaman (2021). It would be helpful to discuss the potential advantages and challenges of using such matrices, and to consider whether incorporating them into the current study could have led to different conclusions or insights.

The manuscript would benefit from careful proofreading to correct minor grammatical errors and ensure consistent formatting throughout.

In conclusion, the manuscript presents a valuable contribution to the literature on the factors influencing the spread of COVID-19 in European regions. I recommend the authors revise the manuscript to address these concerns before resubmitting it for consideration.

Author Response

(The authors gave the same response as above.)

Round 2

Reviewer 1 Report

The authors have responded to all my comments of the previous version. Well done!

My only remaining, minor, comment is that the authors should make sure that all tables and figures also in the appendix are referred to the text. 

Reviewer 2 Report

The present version of the article attends all the issues I identified previously. As such, I think it is ready to be published in its current form.

I leave it to the discretion of the Editor whether the article may exceed the word limit.